# Neural Word Segmentation with Rich Pretraining

## Abstract

Neural word segmentation research has benefited from large-scale raw texts by leveraging them for pretraining character and word embeddings. On the other hand, statistical segmentation research has exploited richer sources of external information, such as punctuation, automatic segmentation and POS. We investigate the effectiveness of a range of external training sources for neural word segmentation by building a modular segmentation model, pretraining the most important sub module using rich external sources. Results show that such pretraining significantly improves the model, leading to accuracies competitive to the best methods on six benchmarks.

## 1   Introduction

There has been a recent shift of research attention in the word segmentation literature from statistical methods to deep learning (Zheng et al., 2013; Pei et al., 2014; Morita et al., 2015; Chen et al., 2015b; Cai and Zhao, 2016; Zhang et al., 2016). Neural network models have been exploited due to their strength in *non-sparse representation learning* and *non-linear power* in feature combination, which have led to advances in many NLP tasks. So far, neural word segmentors have given comparable accuracies to the best statical models.

With respect to *non-sparse representation*, character embeddings have been exploited as a foundation of neural word segmentors. They serve to reduce sparsity of character ngrams, allowing, for example, "猫(cat) 躺(lie) 在(in) 墙角(corner)" to be connected with "狗(dog) 蹲(sit) 在(in) 墙角(corner)" (Zheng et al., 2013), which is infeasible by using sparse one-hot character features. In addition to character embeddings, distributed representations of character bigrams (Mansur et al., 2013; Pei et al., 2014) and words (Morita et al., 2015; Zhang et al., 2016) have also been shown to improve segmentation accuracies.

With respect to *non-linear modeling power*, various network structures have been exploited to represent contexts for segmentation disambiguation, including multi-layer perceptrons on five-character windows (Zheng et al., 2013; Mansur et al., 2013; Pei et al., 2014; Chen et al., 2015a), as well as LSTMs on characters (Chen et al., 2015b; Xu and Sun, 2016) and words (Morita et al., 2015; Cai and Zhao, 2016; Zhang et al., 2016). For structured learning and inference, CRF has been used for character sequence labelling models (Pei et al., 2014; Chen et al., 2015b) and structural beam search has been used for word-based segmentors (Cai and Zhao, 2016; Zhang et al., 2016).

Previous research has shown that segmentation accuracies can be improved by pretraining character and word embeddings over large Chinese texts, which is consistent with findings on other NLP tasks, such as parsing (Andor et al., 2016). Pretraining can be regarded as one way of leveraging external resources to improve accuracies, which is practically highly useful and has become a standard practice in neural NLP. On the other hand, statistical segmentation research has exploited raw texts for semi-supervised learning, by collecting clues from raw texts more thoroughly such as mutual information and punctuation (Li and Sun, 2009; Sun and Xu, 2011), and making use of self-predictions (Wang et al., 2011; Liu and Zhang, 2012). It has also utilised heterogenous annotations such as POS (Ng and Low, 2004; Zhang and Clark, 2008) and segmentation under different standards (Jiang et al., 2009). To our knowledge, such rich external information has not been systematically investigated for neural segmentation.

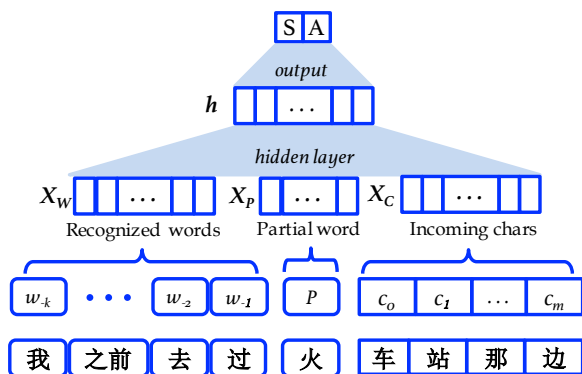

Figure 1: Overall model.

We fill this gap by investigating rich external pretraining for neural segmentation. Following Cai and Zhao (2016) and Zhang et al. (2016), we adopt a globally optimised beam-search framework for neural structured prediction (Andor et al., 2016; Zhou et al., 2015; Wiseman and Rush, 2016), which allows word information to be modelled explicitly. Different from previous work, we make our model conceptually simple and modular, so that the most important sub module, namely a five-character window context, can be pretrained using external data. We adopt a multi-task learning strategy (Collobert et al., 2011), casting each external source of information as a auxiliary classification task, sharing a five-character window network. After pretraining, the character window network is used to initialize the corresponding module in our segmentor.

Results on 6 different benchmarks show that our method outperforms the best statistical and neural segmentation models consistently, giving the best reported results on 5 datasets in different domains and genres. Our code and models can be downloaded from GitHub.com/XXXXXXXXX

## 2 Related Work

Work on *statistical* word segmentation dates back to the 1990s (Sproat et al., 1996). State-of-the-art approaches include *character* sequence labeling models (Xue et al., 2003) using CRFs (Peng et al., 2004; Zhao et al., 2006) and max-margin structured models leveraging *word* features (Zhang and Clark, 2007; Sun et al., 2009; Sun, 2010). Semi-supervised methods have been applied to both character-based and word-based models, exploring external training data for better segmentation (Sun and Xu, 2011; Wang et al., 2011; Liu and

Zhang, 2012; Zhang et al., 2013). Our work belongs to recent *neural* word segmentation.

To our knowledge, there has no work in the literature systematically investigating rich external resources for neural word segmentation training. Closest in spirit to our work, Sun and Xu (2011) empirically studied the use of various external resources for enhancing a statistical segmentor, including character mutual information, access variety information, punctuation and other statistical information. Their baseline is similar to ours in the sense that both character and word contexts are considered. On the other hand, their model is statistical while ours is neural. Consequently, they integrate external knowledge as features, while we integrate it by shared network parameters. Our results show a similar degree of error reduction compared to theirs by using external data.

Our model inherits from previous findings on context representations, such as character windows (Mansur et al., 2013; Pei et al., 2014; Chen et al., 2015a) and LSTMs (Chen et al., 2015b; Xu and Sun, 2016). Similar to Zhang et al. (2016) and Cai and Zhao (2016), we use word context on top of character context. However, words play a relatively less important role in our model, and we find that word LSTM, which has been used by all previous neural segmentation work, is unnecessary for our model. Our model is conceptually simpler and more modularised compared with Zhang et al. (2016) and Cai and Zhao (2016), allowing a central sub module, namely a five-character context window, to be pretrained.

## 3 Model

Our segmentor works incrementally from left to right. At each step, the state consists of a sequence of words that have been fully recognized, denoted as $W = [w_{-k}, w_{-k+1}, ..., w_{-1}]$, a current partially recognized word $P$, and a sequence of next incoming characters, denoted as $C = [c_0, c_1, ..., c_m]$, as shown in Figure 1. Given an input sentence, $W$ and $P$ are initialized to $[\,]$ and $\phi$, respectively, and $C$ contains all the input characters. At each step, a decision is made on $c_0$, either appending it as a part of $P$, or seperating it as the beginning of a new word. The incremental process repeats until $C$ is empty and $P = \phi$ again. Formally, the process can be regarded as a state-transition process, where a state is a tuple $S = \langle W, P, C \rangle$, and the transition actions include SEP and APP, as shown

Axiom: $S = \langle [\,], \phi, C \rangle$, $V = 0$
Goal: $S = \langle W, \phi, [\,] \rangle$, $V = V_{final}$

SEP: $\dfrac{S = \langle W, P, c_0 | C \rangle, V}{S' = \langle W | P, c_0, C \rangle, V + Score(S, \text{SEP})}$

APP: $\dfrac{S = \langle W, P, c_0 | C \rangle, V}{S' = \langle W, P \oplus c_0, C \rangle, V + Score(S, \text{APP})}$

Figure 2: Deduction system, where $\oplus$ denotes string concatenation.

by the deduction system in Figure 2[1].

In the figure, $V$ denotes the score of a state, given by a neural network model. The score of the initial state (i.e. axiom) is 0, and the score of a non-axiom state is the sum of scores of all incremental decisions resulting in the state. Similar to Zhang et al. (2016) and Cai and Zhao (2016), our model is a global structural model, using the overall score to disambiguate states, which correspond to sequences of inter-dependent transition actions.

Different from previous work, the structure of our scoring network is shown in Figure 1. It consists of three main layers. On the bottom is a *representation layer*, which derives dense representations $X_W$, $X_P$ and $X_C$ for $W, P$ and $C$, respectively. We compare various distributed representations and neural network structures for learning $X_W$, $X_P$ and $X_C$, detailed in Section 3.1. On top of the representation layer, we use a *hidden layer* to merge $X_W$, $X_P$ and $X_C$ into a single vector

$$h = tanh(W_{hW} \cdot X_W + W_{hP} \cdot X_P + W_{hC} \cdot X_C + b_h)$$ (1)

The hidden feature vector $h$ is used to represent the state $S = \langle W, P, C \rangle$, for calculating the scores of the next action. In particular, a linear *output layer* with two nodes is employed:

$$o = W_o \cdot h + b_o$$ (2)

The first and second node of $o$ represent the scores of SEP and APP given $S$, namely $Score(S, \text{SEP})$, $Score(S, \text{APP})$ respectively.

### 3.1 Representation Learning

**Characters.** We investigate two different approaches to encoding incoming characters, namely

---

[1]An end of sentence symbol $\langle /s \rangle$ is added to the input so that the last partial word can be put onto $W$ as a full word before segmentation finishes.

a *window approach* and an *LSTM approach*. For the former, we follow prior methods (Xue et al., 2003; Pei et al., 2014), using five-character window $[c_{-2}, c_{-1}, c_0, c_1, c_2]$ to represent incoming characters. Shown in Figure 3, multi-layer perceptron (MLP) is employed to derive a five-character window vector $D_C$ from single-character vector representations $V_{c_{-2}}, V_{c_{-1}}, V_{c_0}, V_{c_1}, V_{c_2}$.

$$D_C = MLP([V_{c_{-2}}; V_{c_{-1}}; V_{c_0}; V_{c_1}; V_{c_2}])$$ (3)

For the latter, we follow recent work (Chen et al., 2015b; Zhang et al., 2016), using a bi-directional LSTM to encode input character sequence.[2] In particular, the bi-directional LSTM hidden vector $[\overleftarrow{h_C}(c_0); \overrightarrow{h_C}(c_0)]$ of the next incoming character $c_0$ is used to represent the coming characters $[c_0, c_1, ...]$ given a state. Intuitively, a five-character window provides a local context from which the meaning of the middle character can be better disambiguated. LSTM, on the other hand, captures larger contexts, which can contain more useful clues for dismbiguation but also irrelevant information. It is therefore interesting to investigate a combination of their strengths, by first deriving a locally-disambiguated version of $c_0$, and then feed it to LSTM for a globally disambiguated representation.

Now with regard to the single-character vector representation $V_{c_i}(i \in [-2, 2])$, we follow previous work and consider both character embedding $e^c(c_i)$ and character-bigram embedding $e^b(c_i, c_{i+1})$, investigating the effect of each on the accuracies. When both $e^c(c_i)$ and $e^b(c_i, c_{i+1})$ are utilized, the concatenated vector is taken as $V_{c_i}$.

**Partial Word.** We take a very simple approach to representing the partial word $P$, by using the embedding vectors of its first and last characters, as well as the embedding of its length. Length embeddings are randomly initialized and then tuned in model training. $X_P$ has relatively less influence on the empirical segmentation accuracies.

$$X_P = [e^c(P[0]); e^c(P[-1]); e^l(\text{LEN}(P))]$$ (4)

**Word.** Similar to the character case, we investigate two different approaches to encoding incoming characters, namely a *window approach* and an *LSTM approach*. For the former, we follow prior

---

[2]The LSTM variation with coupled input and forget gate but without peephole connections is applied (Gers and Schmidhuber, 2000)

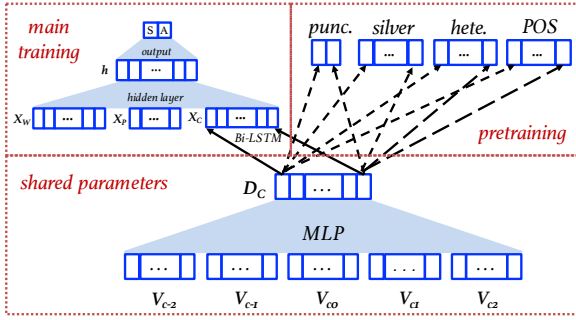

Figure 3: Shared character representation.

methods (Zhang and Clark, 2007; Sun, 2010), using the two-word window $[w_{-2}, w_{-1}]$ to represent recognized words. A hidden layer is employed to derive a two-word vector $X_W$ from single word embeddings $e^w(w_{-2})$ and $e^w(w_{-1})$.

$$X_W = tanh(W_w[e^w(w_{-2}); e^w(w_{-1})] + b_w) \quad (5)$$

For the latter, we follow Zhang et al. (2016) and Cai and Zhao (2016), using an uni-directional LSTM on words that have been recognized.

### 3.2 Pretraining

Neural network models for NLP benefit from pretraining of word/character embeddings, learning distributed sementic information from large raw texts for reducing sparsity. The three basic elements in our neural segmentor, namely characters, character bigrams and words, can all be pretrained over large unsegmented data. We pretrain the five-character window network in Figure 3 as an unit, learning the MLP parameter together with character and bigram embeddings. We consider four types of commonly explored external data to this end, all of which have been studied for statistical word segmentation, but not for neural network segmentors.

**Raw Text.** Although raw texts do not contain explicit word boundary information, statistics such as mutual information between consecutive characters can be useful features for guiding segmentation (Sun and Xu, 2011). For neural segmentation, these distributional statistics can be implicitly learned by pretraining character embeddings. We therefore consider a more explicit clue for pretraining our character window network, namely punctuations (Li and Sun, 2009).

Punctuation can serve as a type of explicit markup (Spitkovsky et al., 2010), indicating that the two characters on its left and right belong to two different words. We leverage this source of information by extracting character five-grams excluding punctuation from raw sentences, using them as inputs to classify whether there is punctuation before middle character. Denoting the resulting five character window as $[c_{-2}, c_{-1}, c_0, c_1, c_2]$, the MLP in Figure 3 is used to derive its representation $D_C$, which is then fed to a softmax layer for binary classification:

$$P(punc) = softmax(W_{punc} \cdot D_C + b_{punc}) \quad (6)$$

Here $P(punc)$ indicates the probability of a punctuation mark existing before $c_0$. Standard back-propagation training of the MLP in Figure 3 can be done jointly with the training of $W_{punc}$ and $b_{punc}$. After such training, the embedding $V_{ci}$ and MLP values can be used to initialize the corresponding parameters for $D_C$ in the main segmentor, before its training.

**Automatically Segmented Text.** Large texts automatically segmented by a baseline segmentor can be used for self-training (Liu and Zhang, 2012) or deriving statistical features (Wang et al., 2011). We adopt a simple strategy, taking automatically segmented text as silver data to pretrain the five-character window network. Given $[c_{-2}, c_{-1}, c_0, c_1.c_2]$, $D_C$ is derived using the MLP in Figure 3, and then used to classify the segmentation of $c_0$ into B(begining)/M(middle)/E(end)/S(single character word) labels.

$$P(silver) = softmax(W_{silv} \cdot D_C + b_{silv}) \quad (7)$$

Here $W_{silv}$ and $b_{silv}$ are model parameters. Training can be done in the same way as training with punctuation.

**Heterogenous Training Data.** Multiple segmentation corpora exist for Chinese, with different segmentation granularities. There has been investigation on leveraging two corpora under different annotation standards to improve statistical segmentation (Jiang et al., 2009). We try to utilize heterogenous treebanks by taking an external treebank as labeled data, training a B/M/E/S classifier for the character windows network.

$$P(hete) = softmax(W_{hete} \cdot D_C + b_{hete}) \quad (8)$$

**POS Data.** Previous research has shown that POS information is closely related to segmentation (Ng

| Paramater | Value | Paramater | Value |
|-----------|-------|-----------|-------|
| $\alpha$ | 0.01 | $size(e^c)$ | 50 |
| $\lambda$ | $10^{-8}$ | $size(e^b)$ | 50 |
| $p$ | 0.2 | $size(e^w)$ | 50 |
| $\eta$ | 0.2 | $size(e^l)$ | 20 |
| MLP layer | 2 | $size(X_C)$ | 150 |
| beam $B$ | 8 | $size(X_P)$ | 50 |
| $size(h)$ | 200 | $size(X_W)$ | 100 |

Table 1: Hyper-parameter values.

and Low, 2004; Zhang and Clark, 2008). We verify the utility of POS information for our segmentor by pretraining a classifier that predicts the POS on each character, according to the character window representation $D_C$. In particular, given $[c_{-2}, c_{-1}, c_0, c_1, c_2]$, the POS of the word that $c_0$ belongs to is used as the output.

$$P(pos) = softmax(W_{pos} \cdot D_C + b_{pos}) \quad (9)$$

**Multitask Learning.** While each type of external training data can offer one source of segmentation information, different external data can be complimentary to each other. We aim to inject all sources of information into the character window representation $D_C$ by using it as a shared representation for different classification tasks. Neural model have been shown capable of doing multi-task learning via parameter sharing (Collobert et al., 2011). Shown in Figure 3, in our case, the output layer for each task is independent, but the hidden layer $D_C$ and all layers below $D_C$ are shared. We randomly sample from different training sources according to their sizes, performing mixed training.

## 4 Decoding and Training

To train the main segmentor, we adopt the global transition-based learning and beam-search strategy of Zhang and Clark (2011). For decoding, standard beam search is used, where the $B$ best partial output hypotheses at each step are maintained in an *agenda*. Initially, the agenda contains only the start state. At each step, all hypotheses in the agenda are expanded, by applying all possible actions and $B$ highest scored resulting hypotheses are used as the agenda for the next step.

For training, the same decoding process is applied to each training example $(x^i, y^i)$. At step $j$, if the gold-standard sequence of transition actions $y_j^i$ falls out of the agenda, max-margin update is

---

**Algorithm 1:** Training

**Input** : $(x^i, y^i)$
**Parameters:** $\Theta$
**Process:**
$agenda \leftarrow (S = \langle[\,], \phi, X^i\rangle, V = 0)$
**for** *j in [0:*LEN*($X^i$)]* **do**
 $beam = [\,]$
 **for** $\hat{y}$ **in** *agenda* **do**
 $\hat{y}' = $ ACTION$(\hat{y}, $ SEP$)$
 ADD$(\hat{y}', beam)$
 $\hat{y}' = $ ACTION$(\hat{y}, $ APP$)$
 ADD$(\hat{y}', beam)$
 **end**
 $agenda \leftarrow $ TOP$(beam, B)$
 **if** $y_j^i \notin agenda$ **then**
 $\hat{y}_j = $ BESTIN$(agenda)$
 UPDATE$(y_j^i, \hat{y}_j, \Theta)$
 **return**
 **end**
**end**
$\hat{y} = $ BESTIN$(agenda)$
UPDATE$(y^i, \hat{y}, \Theta)$
**return**

---

performed by taking the current best hypothesis $\hat{y}_j$ in the beam as a negative example, and $y_j^i$ as a positive example. The loss function is

$$l(\hat{y}_j, y_j^i) = (score(\hat{y}_j) + \eta \cdot \delta(\hat{y}_j, y_j^i)) \\ - score(y_j^i), \quad (10)$$

where $\delta(\hat{y}_j, y_j^i)$ is the number of incorrect local decisions in $\hat{y}_j$, and $\eta$ controls the score margin.

The strategy above is *early-update* (Collins and Roark, 2004). On the other hand, if the gold-standard hypothesis does not fall out of the agenda until the full sentence has been segmented, yet does not score the highest in the agenda after the last step, a final update is made between the highest scored hypothesis $\hat{y}$ in the agenda and the gold-standard $y^i$, using exactly the same loss function. Pseudocode for the online learning algorithm is shown in Algorithm 1.

We use *Adagrad* (Duchi et al., 2011) to optimize model parameters, with an initial learning rate $\alpha$. $L2$ regularization and dropout (Srivastava et al., 2014) on input are used to reduce overfitting, with a $L2$ weight $\lambda$ and a dropout rate $p$. All the parameters in our model are randomly initialized to a value $(-r, r)$, where $r = \sqrt{\frac{6.0}{fan_{in} + fan_{out}}}$ (Bengio, 2012). We fine-tune character and character

|  | Source | #Chars | #Words | #Sents |
|---|---|---|---|---|
| Raw data | Gigaword | 116.5m | – | – |
| Auto seg | Gigaword | 398.2m | 238.6m | 12.04m |
| Hete. | People's Daily | 10.14m | 6.17m | 104k |
| POS | People's Daily | 10.14m | 6.17m | 104k |

Table 2: Statistics of external data.

bigram embeddings, but not word embeddings, acccording to Zhang et al. (2016).

## 5 Experiments

### 5.1 Experimental Settings

**Data.** We use Chinese Treebank 6.0 (CTB6) (Xue et al., 2005) as our main dataset. Training, development and test set splits follow previous work (Zhang et al., 2014). In order to verify the robustness of our model, we additionally use SIGHAN 2005 bake-off (Emerson, 2005) and NLPCC 2016 shared task for Weibo segmentation (Qiu et al., 2016) as test datasets, where the standard splits are used. For pretraining embedding of words, characters and character bigrams, we use Chinese Gigaword (simplified Chinese sections)[3], automatically segmented using ZPar 0.6 off-the-shelf (Zhang and Clark, 2007), the statictics of which are shown in Table 2.

For pretraining character representations, we extract punctuation classification data from the Gigaword corpus, and use the word-based ZPar and a standard character-based CRF model (Tseng et al., 2005) to obtain automatic segmentation results. We compare pretraining using ZPar results only and using results that both segmentors agree on. For heterogenous segmentation corpus and POS data, we use a People's Daily corpus of 5 months[4]. Statistics are listed in Table 2.

**Evaluation.** The standard word precision, recall and F1 measure (Emerson, 2005) are used to evaluate segmentation performances.

**Hyper-parameter Values.** We adopt commonly used values for most hyperparameters, but tuned the sizes of hidden layers on the development set. The values are summarized in Table 1.

### 5.2 Development Experiments

We perform development experiments to verify the usefulness of various context representations, network configurations and different pretraining methods, respectively.

---

[3]https://catalog.ldc.upenn.edu/LDC2011T13
[4]http://www.icl.pku.edu.cn/icl_res

| Character | P | R | F |
|---|---|---|---|
| No char | 82.19 | 87.20 | 84.62 |
| 5-char window | 95.33 | 95.50 | 95.41 |
| char LSTM | 95.21 | 95.82 | 95.51 |
| 5-char window+LSTM | 95.77 | 95.95 | **95.86** |
| -char emb | 95.20 | 95.19 | 95.20 |
| -bichar emb | 93.87 | 94.67 | 94.27 |

Table 3: Influence of character contexts.

#### 5.2.1 Context Representations

The influence of character and word context representations are empirically studied by varying the network structures for $X_C$ and $X_W$ in Figure 1, respectively. All the experiments in this section are performed using a beam size of 8.

**Character Context.** We fix the word representation $X_W$ to a 2-word window and compare different character context representations. The results are shown in Table 3, where "no char" represents our model without $X_C$, "5-char window" represents a five-character window context, "char LSTM" represents character LSTM context and "5-char window + LSTM" represents a combination, detailed in Section 3.1. "-char emb" and "-bichar emb" represent the combined window and LSTM context without character and character-bigram information, respectively.

As can be seen from the table, without character information, the F-score is 84.62%, demonstrating the necessity of character contexts. Using window and LSTM representations, the F-scores increase to 95.41% and 95.51%, respectively. A combination of the two lead to further improvement, showing that local and global character contexts are indeed complementary, as hypothesized in Section 3.1. Finally, by removing character and character-bigram embeddings, the F-score decreases to 95.20% and 94.27%, respectively, which suggests that character bigrams are more useful compared to character unigrams. This is likely because they contain more distinct tokens and hence offer a larger parameter space.

**Word Context.** The influence of various word contexts are shown in Table 4. Without using word information, our segmentor gives an F-score of 95.66% on the development data. Using a context of only $w_{-1}$ (1-word window), the F-measure increases to 95.78%. This shows that word contexts are far less important in our model compared to character contexts, and also compared to word contexts in previous word-based segmentors

| Word | P | R | F |
|------|-----|-----|-----|
| No word | 95.50 | 95.83 | 95.66 |
| 1-word window | 95.70 | 95.85 | 95.78 |
| 2-word window | 95.77 | 95.95 | **95.86** |
| 3-word window | 95.80 | 95.85 | 95.83 |
| word LSTM | 95.71 | 95.97 | 95.84 |
| 2-word window+LSTM | 95.74 | 95.95 | 95.84 |

Table 4: Influence of word contexts.

(Zhang et al., 2016; Cai and Zhao, 2016). This is likely due to the difference in our neural network structures, and that we fine-tune both character and character bigram embeddings, which significantly enlarges the adjustable parameter space as compared with Zhang et al. (2016). The fact that word contexts can contribute relatively less than characters in a word is also not surprising in the sense that word-based neural segmentors do not outperform the best character-based models by large margins. Given that character context is what we pretrain, our model relies more heavily on them.

With both $w_{-2}$ and $w_{-1}$ being used for the context, the F-score further increases to 95.86%, showing that a 2-word window is useful by offering more contextual information. On the other hand, when $w_{-3}$ is also considered, the F-score does not improve further. This is consistent with previous findings of statistical word segmentation (Zhang and Clark, 2007), which adopt a 2-word context. Interestingly, using a word LSTM does not bring further improvements, even when it is combined with a window context. This suggests that global word contexts may not offer crucial additional information compared with local word contexts. Intuitively, words are significantly less polysemous compared with characters, and hence can serve as effective contexts even if used locally, to supplement a more crucial character context.

### 5.2.2 Stuctured Learning and Inference

We verify the effectiveness of structured learning and inference by measuring the influence of beam size on the baseline segmentor. Figure 4 shows the F-scores against different numbers of training iterations with beam size 1,2,4,8 and 16, respectively. When the beam size is 1, the inference is local and greedy. As the size of the beam increases, more global structural ambiguities can be resolved since learning is designed to guide search. A contrast between beam sizes 1 and 2 demonstrates the usefulness of structured learning and inference. As

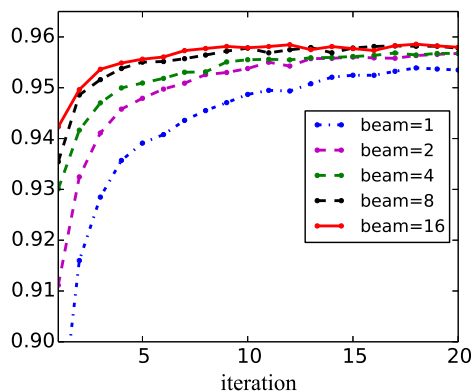

Figure 4: F1 measure against the training epoch.

| Pretrain | P | R | F | ER% |
|----------|-----|-----|-----|-----|
| Baseline | 95.77 | 95.95 | 95.86 | 0 |
| +Punc. pretrain | 96.36 | 96.13 | 96.25 | -9.4 |
| +Auto-seg pretrain | 96.23 | 96.29 | 96.26 | -9.7 |
| +Heter-seg pretrain | 96.28 | 96.27 | 96.27 | -9.9 |
| +POS pretrain | 96.16 | 96.28 | 96.22 | -8.7 |
| +Multitask pretrain | 96.49 | 96.39 | 96.44 | -14.0 |

Table 5: Influence of pretraining.

the beam size increases, the gain by doubling the beam size decreases. We choose a beam size of 8 for the remaining experiments for a tradeoff between speeds and accuraces.

### 5.2.3 Pretraining Results

Table 5 shows the effectiveness of rich pretraining of $D_c$ on the development set. In particular, by using punctuation information, the F-score increases from 95.86% to 96.25%, with a relative error reduction of 9.4%. This is consistent with the observation of Sun and Xu (2011), who show that punctuation is more effective compared with mutual information and access variety as semi-supervised data for a statistical word segmentation model. With automatically-segmented data[5], heterogenous segmentation and POS information, the F-score increases to 96.26%, 96.27% and 96.22%, respectively, showing the relevance of all information sources to neural segmentation, which is consistent with observations made for statistical word segmentation (Jiang et al., 2009; Wang et al., 2011; Zhang et al., 2013). Finally, by integrating all above information via multi-task learning, the F-score is further improved to 96.44%, with a 14% relative error reduction.

---

[5]By using ZPar alone, the auto-segmented result is 96.02%, less than using results by matching ZPar and the CRF segmentor outputs.

| Models | P | R | F |
|---|---|---|---|
| Baseline | 95.3 | 95.5 | 95.4 |
| Multitask pretrain | **96.2** | **96.0** | **96.1** |
| Sun and Xu (2011) baseline | 95.2 | 94.9 | 95.1 |
| Sun and Xu (2011) multi-source semi | 95.9 | 95.6 | 95.7 |
| Zhang et al. (2016) neural | 95.3 | 94.7 | 95.0 |
| Zhang et al. (2016)* hybrid | 96.1 | 95.8 | 96.0 |
| Chen et al. (2015a) window | 95.7 | 95.8 | 95.8 |
| Chen et al. (2015b) char LSTM | **96.2** | 95.8 | 96.0 |
| Zhang et al. (2014) POS and syntax | – | – | 95.7 |
| Wang et al. (2011) statistical semi | 95.8 | 95.8 | 95.8 |
| Zhang and Clark (2011) statistical | 95.5 | 94.8 | 95.1 |

Table 6: Main results on CTB6.

| F1 measure | PKU | MSR | AS | CityU | Weibo |
|---|---|---|---|---|---|
| Multitask pretrain | **96.2** | 97.3 | **95.6** | **96.7** | **95.4** |
| Cai and Zhao (2016) | 95.5 | 96.5 | – | – | – |
| Zhang et al. (2016) | 95.1 | 97.0 | – | – | – |
| Zhang et al. (2016)* | 95.7 | **97.7** | – | – | – |
| Pei et al. (2014) | 95.2 | 97.2 | – | – | – |
| Sun et al. (2012) | 95.4 | 97.4 | – | – | – |
| Zhang and Clark (2007) | 94.5 | 97.2 | 94.6 | 95.1 | – |
| Zhang et al. (2006) | 95.1 | 97.1 | 95.1 | 95.1 | – |
| Sun et al. (2009) | 95.2 | 97.3 | – | 94.6 | – |
| Sun (2010) | 95.2 | 96.9 | 95.2 | 95.6 | – |
| Wang et al. (2014) | 95.3 | 97.4 | 95.4 | 94.7 | – |
| Xia et al. (2016) | – | – | – | – | 95.4 |

Table 7: Main results on other test datasets.

## 5.3 Final Results

Our final results on CTB6 are shown in Table 6, which lists the results of several current state-of-the-art methods. Without multitask pretraining, our model gives an F-score of 95.44%, which is higher than the neural segmentor of Zhang et al. (2016), which gives the best accuracies among pure neural segments on this dataset. By using multitask pretraining, the result increases to 96.09%, with a relative error reduction of 14.5%. In comparison, Sun and Xu (2011) investigated heterogenous semi-supervised learning on a state-of-the-art *statistical* model, obtaining a relative error reduction of 13.8%. Our findings show that external data can be as useful for *neural* segmentation as for statistical segmentation.

Our final results compares favourably to the best statistical models, including those using semi-supervised learning (Sun and Xu, 2011; Wang et al., 2011), and those leveraging joint POS and syntactic information (Zhang et al., 2014). In addition, it also outperforms the best neural models, in particular Zhang et al. (2016)*, which is a hybrid neural and statistical model, integrating manual discrete features into their word-based neural model. We achieve the best reported F-score on this dataset. To our knowledge, this is the first time a pure neural network model outperforms all existing methods on this dataset, allowing the use of external data.[6]

In addition to CTB6, which has been the most commonly adopted by recent segmentation research, we additionally evaluate our results on the

SIGHAN 2005 bakeoff and Weibo datasets, to examine cross domain robustness. Different state-of-the-art methods for which results are recorded on these datasets are listed in Table 7. Most neural models reported results only on the PKU and MSR datasets of the bakeoff test sets, which are in simplified Chinese. The AS and CityU corpora are in traditional Chinese, sourced from Taiwan and Hong Kong corpora, respectively. We map them into simplified Chinese before segmentation. The Weibo corpus is in a yet different genre, being social media text. Xia et al. (2016) achieved the best results on this dataset by using a statistical model with features learned using external lexicons, the CTB7 corpus and the People Daily corpus. Similar to Table 6, our method gives the best accuracies on all corpora except for MSR, where it underperforms the hybrid model of Zhang et al. (2016) by 0.4%. To our knowledge, we are the first to report results for a neural segmentor on more than 3 datasets, with competitive results consistently. It verifies that knowledge learned from a certain set of resources can be used to enhance cross-domain robustness in training a neural segmentor for different datasets, which is of practical importance.

## 6 Conclusion

We investigated rich external resources for enhancing neural word segmentation, by building a globally optimised beam-search model that leverages both character and word contexts. Taking each type of external resource as an auxiliary classification task, we use neural multi-task learning to pre-train a set of shared parameters for character contexts. Results show that rich pretraining leads to 14.5% relative error reduction, and our model gives results highly competitive to the best systems on six different benchmarks.

---

[6] We did not investigate the use of lexicons (Chen et al., 2015a,b) in our research, since lexicons might cover different OOV in the training and test data, and hence directly affecting the accuracies, which makes it relatively difficult to compare different methods fairly unless a single lexicon is used for all methods, as observed by Cai and Zhao (2016).

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
