# Peer review of "Neural Word Segmentation with Rich Pretraining"

_ACL 2017 — decision unknown_

[Official Review · Reviewer 1 · rating 3 · confidence 4]
soundness 5 · originality 5 · clarity 4 · impact 3 · substance 4 · appropriateness 5 · meaningful comparison 3 · presentation format Oral Presentation

- Strengths:
i. Well organized and easy to understand
ii. Provides detailed comparisons under various experimental settings and shows
the state-of-the-art performances

- Weaknesses:
i. In experiments, this paper compares previous supervised approaches, but the
proposed method is the semi-supervised approach even if the training data is
enough to train.

- General Discussion:
This paper adopts a pre-training approach to improve Chinese word segmentation.
Based on the transition-based neural word segmentation, this paper aims to
pre-train incoming characters with external resources (punctuation, soft
segmentation, POS, and heterogeneous training data) through multi-task
learning. That is, this paper casts each external source as an auxiliary
classification task. The experimental results show that the proposed method
achieves the state-of-the-art performances in six out of seven datasets. 

This paper is well-written and easy to understand. A number of experiments
prove the effectiveness of the proposed method. However, there exist an issue
in this paper. The proposed method is a semi-supervised learning that uses
external resources to pre-train the characters. Furthermore, this paper uses
another heterogeneous training datasets even if it uses the datasets only for
pre-training. Nevertheless, the baselines in the experiments are based on
supervised learning. In general, the performance of semi-supervised learning is
better than that of supervised learning because semi-supervised learning makes
use of plentiful auxiliary information. In the experiments, this paper should
have compared the proposed method with semi-supervised approaches.

POST AUTHOR RESPONSE

What the reviewer concerned is that this paper used additional
“gold-labeled” dataset to pretrain the character embeddings. Some baselines
in the experiments used label information, where the labels are predicted
automatically by their base models as the authors pointed out. When insisting
superiority of a method, all circumstances should be same. Thus, even if the
gold dataset isn’t used to train the segmentation model directly, it seems to
me that it is an unfair comparison because the proposed method used another
“gold” dataset to train the character embeddings.